# Intrinsically Fluorescent Anti-Cancer Drugs

**DOI:** 10.3390/biology11081135

**Published:** 2022-07-28

**Authors:** Md. Lutful Kabir, Feng Wang, Andrew H. A. Clayton

**Affiliations:** 1Optical Sciences Centre, Department of Physics and Astronomy, School of Science, Computing and Engineering Technologies, Swinburne University of Technology, Melbourne, VIC 3122, Australia; mdlutfulkabir@swin.edu.au; 2Department of Chemistry and Biotechnology, School of Science, Computing and Engineering Technologies, Swinburne University of Technology, Melbourne, VIC 3122, Australia; fwang@swin.edu.au

**Keywords:** cancer, anti-cancer drugs, tyrosine kinase, tyrosine kinase inhibitors, fluorescence

## Abstract

**Simple Summary:**

Cancer is one of the biggest causes of death world-wide. The development of anti-cancer drugs is important in combatting this disease. As good as these drugs appear to be, they do suffer from problems such as side-effects and disease resistance. We surmise that an improved understanding of drug-target and drug non-target interactions is important in addressing these issues. Here we review the use of a spectroscopic method called fluorescence to provide this needed information on target and non-target interactions. Fortunately, some of the drugs used pre-clinically or in the clinic are intrinsically fluorescent and thus can be used as spectroscopic probes of their own environment.

**Abstract:**

At present, about one-third of the total protein targets in the pharmaceutical research sector are kinase-based. While kinases have been attractive targets to combat many diseases, including cancer, selective kinase inhibition has been challenging, because of the high degree of structural homology in the active site where many kinase inhibitors bind. Despite efficacy as cancer drugs, kinase inhibitors can exhibit limited target specificity and rationalizing their target profiles in the context of precise molecular mechanisms or rearrangements is a major challenge for the field. Spectroscopic approaches such as infrared, Raman, NMR and fluorescence have the potential to provide significant insights into drug-target and drug-non-target interactions because of sensitivity to molecular environment. This review places a spotlight on the significance of fluorescence for extracting information related to structural properties, discovery of hidden conformers in solution and in target-bound state, binding properties (e.g., location of binding sites, hydrogen-bonding, hydrophobicity), kinetics as well as dynamics of kinase inhibitors. It is concluded that the information gleaned from an understanding of the intrinsic fluorescence from these classes of drugs may aid in the development of future drugs with improved side-effects and less disease resistance.

## 1. Introduction

The purpose of this review is to highlight the intrinsic fluorescence properties of anti-cancer drugs and the application of fluorescence to study the interaction of these drugs with biomolecules. In Section 2, we provide an introduction to protein tyrosine kinase inhibitors. The purpose of Section 2 is to give the uninitiated an appreciation of this class of drug for cancer therapy. In Section 3.1, we introduce the reader to the basics of fluorescence spectroscopy, so that the meaning of intrinsically fluorescent anti-cancer drugs can be placed in context. In Section 3.2, we narrow down to the quinazoline-based tyrosine-kinase inhibitors, which form the basis for many anti-cancer drugs used in the clinic and for which the quinazoline group not only provides a useful scaffold for drug development but also a convenient chromophore with intrinsic fluorescence. Accordingly, in Section 3.3 we review the basic spectroscopy of these tyrosine kinase inhibitors, revealing their prominent sensitivity to environment. Section 3.4 and subsections then review the use of fluorescence to determine binding to different classes of biomolecules including proteins (Section 3.4.1), DNA (Section 3.4.2), lipids (Section 3.4.3) and cells (Section 3.4.4). Future perspectives in preclinical models for cancer research are briefly noted in Section 3.4.5. We conclude in Section 4.

## 2. Protein Kinase Inhibitors

Before discussing the specifics of intrinsically fluorescent anti-cancer drugs, we first wish to provide an introduction to one of the most sought-after drug targets and their cognate inhibitors, called protein kinase inhibitors. Readers who are already familiar with protein kinase inhibitors may wish to skip this section (Section 2) and move to Section 3. Protein kinases are attractive targets for cancer therapy due to their central role in mediating cellular growth pathways [1]. Today, numerous adenosine triphosphate (ATP)-competitive kinase inhibitors are available that inhibit mutated or over-expressed kinases responsible for driving oncogenic signaling. Kinase inhibitors with improved selectivity are highly desirable, both as new cancer therapeutics and as reagents for studying signaling pathways [2]. Efforts to profile prospective kinase inhibitors against a panel of kinases have revealed that candidate drug compounds can bind to multiple off-target kinases including the intended target kinase [3]. Understanding the origin of off-target binding patterns in terms of molecular mechanism is an important goal that would enhance the use of existing inhibitors and greatly benefit the process of inhibitor development.

Since the Food and Drug Administration (FDA) approval of Imatinib, a kinase inhibitor, in 2001, drugs targeting kinases now account for nearly 50% of current cancer therapeutic discovery efforts [3]. Several tyrosine kinase inhibitors (TKI) have been approved for cancer treatment including those that target the epidermal growth factor receptor (EGFR, also referred as erbB1, Her1). The EGFRs represent a subfamily of tyrosine kinase receptors including Her1, Her2, Her3 and Her4 (also referred to as erbB1–erbB4). Receptor overexpression, receptor mutation, or disruption of signaling from this receptor network are thought to contribute to oncogenesis in various cancer types [4,5]. Gefitinib (GEF, Iressa^®^ AstraZeneca UK Limited, Chesire, England), was the first selective TKI approved in 2003 for the treatment of advanced ormetastatic non-small-cell lung cancer (NSCLC). Erlotinib (ERL, Tarceva^®^, Roche, Basel, Switzerland). Worldwide is a reversible inhibitor and targets the EGFR–ATP binding site. In the clinic, Erlotinib is used to treat EGFR-mutant NSCLC and pancreatic cancer [6]. Afatinib (AFA, Gilotrif^®^, Boehringer Ingelheim Pharmaceuticals, Ingelheim, Germany, FDA-approved 2013) was the first irreversible inhibitor of EGFR, with similar indications as the former two compounds. Unfortunately, these drugs required further development because cancer patients would frequently become resistant to these first- and second-generation inhibitors. The third-generation inhibitor Osimertinib (OSI, Tagrisso^®^, AstraZeneca UK Limited, Chesire, England, approved in 2017) has been specifically developed for cancers driven by the T790M EGFR mutation [7]. Lapatinib is a quinazoline derivative used in the treatment of ErbB2-overexpressing breast cancer. 

EGFR inhibitors are often orally administered and enter the circulation via the bloodstream. As a result, the impact of plasma protein binding is of high importance on drug pharmacodynamics (i.e., drug efficacy, distribution, or elimination) [8]. Notably, plasma half-lives of the EGFR inhibitors GEF, ERL and AFA in patients range from 1.5 to 2 days and extensive plasma protein binding (>90%) was reported [9,10,11], whereas for OSI, high plasma protein binding was assumed only [12]. Thus, thorough investigation of the nature and primary binding sites on the plasma protein might represent an important issue for the understanding and comparison of the pharmacological activity and the pharmacokinetic behavior of these therapeutic agents.

Aside from small molecule inhibitors, which target the kinase domain of one of the EGFR family members, there are also therapeutic anti-bodies, which are directed towards the extracellular domain of the receptor. Trastuzumab, pertuzumab, and adotrastuzumab emtansine, are used for the treatment of ErbB2-positive breast cancer; adotrastuzumab emtansine is an antibody–drug conjugate that delivers a cytotoxic drug to cells overexpressing ErbB2. Cetuximab and panitumumab are monoclonal antibodies that target ErbB1 and are used in the treatment of colorectal cancer, and head and neck cancers. As discussed above, cancers treated with these targeted drugs eventually become resistant to them. Pre-clinical research efforts are being focused on the so-called combination therapies, which involve different drug combinations (e.g., small molecule drug with antibody therapy) or combinations of targeted drugs with cytotoxic therapies. Combination therapies hold promise as one route to preventing or delaying the onset of drug resistance and patient relapse [13].

## 3. Fluorescence, Tyrosine-Kinase Inhibitors and Beyond

### 3.1. What Is Fluorescence?

In order for us to appreciate intrinsically fluorescent anti-cancer drugs we first introduce the reader to the basics of fluorescence itself [14]. Fluorescence, a form of luminescence that can occur either in gas, liquid or solid chemical systems, is the emission of electromagnetic radiation, usually visible light, by a material that has absorbed light or other electromagnetic radiation. In the case of fluorescence, as shown in Figure 1a, the emitted light has a longer wavelength, i.e., lower photon energy compared to the absorbed radiation. This difference in wavelength between the positions of those band maxima is called the Stokes shift.

The mechanism of fluorescence can be explained with the help of the Jablonski energy diagram (Figure 1b). The upward violet arrow represents the absorption of a photon in the singlet electronic ground state (S_o_), causing a promotion of an electron to the singlet excited electronic state, S_1_. The downward red arrows denote vibrational relaxation from vibrationally excited states within the S_1_ manifold. This is a non-radiative relaxation process (i.e., no photon is emitted) in this case because the excitation energy is dispersed as vibrations or heat to the solvent. The downward green arrow denotes the fluorescence process from S_1_ to S_0_.

Note that because fluorescence occurs from the lowest vibrational level of the excited-state (S_1_) to higher vibrational levels of the ground-state (S_0_), the emission maximum of the fluorescence spectrum is always at lower energy (longer wavelength) than the excitation.

The quantum yield is an important parameter in fluorescence and its sensitivity to environment forms the basis for assays for detecting drug binding to drug targets. The quantum yield is defined as the number of photons emitted per photon absorbed and is in the range of 0 to 1. The quantum yield is determined by the relative rates of non-radiative and radiative processes that deplete the excited state. Radiative and non-radiative rates can be very dependent on environment, such as the binding sites of drug targets, so drug binding can be accompanied by large changes in quantum yield (or equivalently changes in amplitude of the fluorescence spectrum).

Another important fluorescence parameter is called the fluorescence lifetime. The fluorescence lifetime is the average time a molecule spends in the excited state. In contrast to quantum yield, which is related to relative rates of processes, the fluorescence lifetime is defined as the reciprocal of the sum of all the rates of processes depleting the excited state. Typical fluorescence lifetimes are in the range of 1–10 ns.

The finite lifetime of the excited state means that fluorescence can be sensitive to molecular structure and dynamics in the vicinity of the fluorophore. Myriad processes can occur during the excited state including rotation, collisions with other molecules, or a change in structure or conformation and environmental (solvent) relaxation. 

Solutions containing the fluorophore are normally studied with a special spectrometer called a fluorometer, usually with a single exciting wavelength and variable detection wavelength (scanned to create a fluorescence spectrum). Because of the sensitivity that the method affords, fluorescent molecule concentrations as low as the nanomolar level can be measured [14]. Fluorescence in several wavelengths can be detected by an array detector, to detect compounds from high performance liquid chromatography (HPLC) flow. Thin layer chromatography (TLC) plates can also be visualized if the compounds or a coloring reagent is fluorescent. For visualizing fluorescence in cells, one uses a fluorescence microscope.

Molecular structure and chemical environment affect whether or not a substance undergoes fluorescence. When fluorescence does occur, molecular structure and local environment determine the color and intensity of emission. Hence, fluorescence can be used to investigate binding affinities, binding mechanisms, properties (i.e., polarity) of the binding site on the protein and binding kinetics. Another example is the effects of solvent on the structure and spectroscopic behavior of a fluorophore. Generally, solvatochromism is observed due to the differential solvation of the ground and excited states of a fluorophore. It was found that the optical spectroscopic measurements of a fluorophore can be influenced by the change in physicochemical properties of the surrounding medium. Solvatochromism is the term used to define this phenomenon and firstly introduced by Hantzschlater. The change in compound absorption/emission spectrum is manifested by one or more alternations in the band position, intensity or shape [15,16,17]. The hypsochromic (blue) shift of the fluorescence band relative to the absorption band is commonly known as negative solvatochromism. While positive solvatochromism is the term given for the bathochromic (red) shift of the fluorescence band [18], in negative solvatochromism, the molecule in its ground state is more stabilized than in the excited state upon increasing solvent polarity. When the excited state is more stabilized than the ground state, it results in a positive solvatochromism [18].

Generally, molecules that fluoresce are conjugated systems. Since most of the tyrosine kinase inhibitors are small aromatic molecules, their extended conjugation and ionizable groups render them good candidates for intrinsically fluorescent anti-cancer drugs. 

### 3.2. Quinazoline-Based TKI

Quinazoline is one of the most widespread scaffolds among natural and synthetic bioactive compounds. Quinazoline scaffold resembles both the purine nucleus and the pteridine one. As a consequence, some compounds able to inhibit the purinic [19] or the folic acid [20] metabolic pathways have been discovered. Over the past few decades, the therapeutic potential of quinazoline derivatives has been found as different types of anticancer agents such as protein kinase inhibitors, tubulin polymerization inhibitors, protein lysine methyltransferase inhibitors, topoisomerase I inhibitors, PI3K/Akt/mTOR inhibitors, poly(ADP-ribose)polymerase-1 (PARP-1) inhibitors etc. [21]. Quinazoline ring is very commonly found in all types of TKIs [22]. In general, quinazoline derivatives are known to possess a wide range of activities. A specific activity depends on the substituent present at an appropriate position of quinazoline.

For designing TKIs, 4-anilinoquinazolines found to be potent selective inhibitors [23]. The 4-anilinoquinazoline scaffold, i.e., the quinazoline core and N-aryl arm (shown in Figure 2), together, form established ErbB/EGFR pharmacophore and used in type I (e.g., Gefitinib), type II (e.g., Lapatinib) and covalent inhibitors (e.g., Dacomitinib). This pharmacophore N-aryl arm is oriented deep for binding to the ATP nucleotide pocket [24]. The introduction of strong electron-donating or -withdrawing groups to the tail section (shown in Figure 2) being projected into the solvent produces donor–acceptor systems that are extremely sensitive to solvent polarity. As Figure 2 shows, to increase the selectivity and potency of TKIs, 4-phenylaminooquinazolines are substituted at different positions with suitable substituent groups/atoms [25]. The first- and second-generation EGFR–TKIs both have aniline-quinazoline structures. However, the second-generation TKIs also have an acrylamide group at 6 position of quinazoline ring (see Figure 2), which serves as a chemically reactive electrophile called a Michael acceptor [26] that targets a cysteine nucleophile (Cys-797), resulting in a covalent adduct. Importance of fluorophore arm and pharmacophore arm of the 4-aminoquinazolines is well documented by J. Dhuguru et al. [24].

### 3.3. Basic Fluorescence Spectroscopy of TKIs

Solvatochromism is commonly used in many research aspects to characterize the nature of bulk environment [27]. Studying solvent effects on a fluorophore is commonly utilized to estimate the photophysical properties of a fluorophore. Fluorophore interaction with its environment can be triggered through specific interactions between fluorophore and solvent molecules, e.g., H-bond formation and non-specific interactions. Electrostatic interactions, driven by the change in solvent dipole moment, refractive index, relative permittivity (or dielectric constant), polarizability and viscosity, can induce a significant change in fluorophore electronic configuration and spectrum [28]. 

A number of empirical solvent models have been developed for the solvatochromic analysis of organic molecules. Linear solvation energy relationship (LSER) models have been proven to reliably define the interactions between solute and solvent molecules. Models developed by Bilot–Kawski [29] Lippert-Abboud–Mataga (L–M) [30,31], Bakhshiev [32] and Kawski–Chamma–Viallet [33] are among the most commonly used LSER models for solvatochromism analysis.

A summary of selected spectroscopic properties of TKI’s is collected in Table 1 [34,35,36,37,38,39,40,41,42,43,44,45,46,47,48,49]. 

The spectroscopy properties depend on the type of drug and also the solvent in which the spectrum is measured. In general, for all of the drugs listed, the absorption transitions all occur in the ultraviolet region of the spectrum. The emission spectra are somewhat more sensitive to drug type and particularly sensitive to environment. Specific examples are illustrated below.

Khattab et al. [50] have carried out a detailed examination of the absorption and fluorescence spectra of AG1478 in twenty-one solvents of different polarity and hydrogen-bonding strength. Solvatochromic analyses using different solvent models was undertaken to investigate the potential of AG1478 as a fluorescent reporter of its own environment (see Figure 3). Fluorescence spectral analyses showed that hydrogen bonding from the solvent to AG1478 plays important role in solvatochromism, showing synergistic effect with solvent polarity in stabilizing the excited state. Fluorescence quantum yields were found to be influenced by solvent H-bond donor ability, being higher in aprotic than in protic solvents. This work underscores the importance of polarity and hydrogen bonding environment on the spectroscopic properties of AG-1478 as well as helps in understanding the interaction of AG-1478 in vitro and in vivo. Moreover, AG-1478 shares the same molecular scaffold with the commercial anti-cancer drugs Erlotinib and Gefitinib [51,52].

Water molecules are frequently found in protein binding sites and can be important determinants for drug–protein intermolecular interactions and stability. Therefore, binding interactions between explicit water molecules and TKI may influence the nature and diversity of drug chemical structures and properties. To better understand the importance of hydration bonding and the specific solvation by water molecules, Khattab et al. [53] used fluorescence spectroscopy in acetonitrile–water solutions. A significant quenching of the AG1478 fluorescence with added water was observed and attributed to the formation of specific complexes between AG1478 and water molecules. Using a combination of computational chemistry with the spectroscopic measurements, the authors identified three potential sites for H-bond formation between AG1478 and water molecules. The computational models predicted that the extent of hydrogen bonding to the drug affected the distribution of twisted and planar drug conformations. A possible but yet to be further proven explanation of the water-induced fluorescence quenching is that the water induces a twisted form of the drug, which is non-fluorescent. As discussed next in this review, the environmental sensitivity of some of these drugs makes them useful reporters when applied to drug–protein complexes. For example, fluorescence when applied to drug–protein complexes can be used (in combination with computational chemistry) to infer drug conformation (from the fluorescence excitation spectrum) and deduce the polarity of the binding site (the fluorescence emission spectrum).

### 3.4. Fluorescence Binding Assay

Fluorescence spectroscopy is a sensitive approach for measuring inhibitor binding interactions with proteins, membranes and DNA. A given change in inhibitor fluorescence intensity (or emission color) upon binding affords a convenient method for quantifying inhibitor binding, which can be used to measure the binding constants of a TKI inhibitor bound to a protein, membrane or DNA [14]. The way in which the binding assay can be employed is as follows. First, the fluorescence (spectrum or intensity at fixed emission wavelength) from the free drug in solution is measured. Then, the binding partner is titrated into the solution and the fluorescence is subsequently measured. Because DNA, proteins and lipids do not absorb light at the excitation wavelength typically used to excite inhibitors, only the emissions from the inhibitor (in free and bound states) are being recorded. Provided the inhibitor concentration is kept constant during the titration and the partner molecule concentration is increased, inhibitor binding will be detected by a change in the fluorescence (increase or decrease). A plot of the fluorescence as a function of the concentration of added molecular partner is called a binding curve. The binding curve can then be analyzed by various models to determine equilibrium binding constants. Another approach to binding uses a phenomenon called fluorescence polarization or anisotropy [54,55]. This assay measures the rotational diffusion of a small probe attached or intrinsic to the drug. Binding to the larger protein, slows down the rotational diffusion and causes an increase in the polarization of the emission from the drug. Conversely competition assays with unlabeled drug, cause a decrease in polarization due to release of the fluorescent drug. The latter approach is very amenable to a high throughput screening approach.

#### 3.4.1. Fluorescence Analysis of Drug–Protein Interactions

Fluorescence spectroscopy is a useful technique to investigate drug–protein interactions. This is because drug fluorescence is very sensitive to the microenvironment; thus, the amplitude or intensity (quantum yield), color (wavelength position), and kinetics (excited-state lifetime) may be strongly affected by the surroundings of the investigated chromophore [56]. With judicious application of sophisticated time-resolved technology, fluorescence can reveal the nature of the early primary processes occurring from the initially excited states over a broad dynamic range (from the femto-second to the nano-second range), such as energy transfer, charge transfer and intersystem crossing. Thus, drug conformation and drug–protein interactions in the vicinity of the amino acid residues of the protein binding sites can be investigated in detail [57,58,59,60]. For instance, the research group of Miranda et al. used both steady-state and time-resolved fluorescence spectroscopy to study the structural and dynamic features of Lapatinib-protein complexation [61]. These results provided important information on the nature of the binding sites and the conformational rearrangements of Lapatinib within the protein cavities. An important aspect was the complementation of the experimental results with simulation studies. In another study the same group [62] examined the interaction of Lapitinib and its metabolites with serum proteins. The fluorescence results pointed to a higher affinity of Lapatinib and its metabolites to human proteins (human serum albumin (HSA)) in comparison to bovine proteins; the highest affinity complex was found for Lapatinib-HSA. These results are relevant to understanding the pharmacodynamics of these drugs.

The plasticity of structural elements of the ATP-binding site is critical for determining the inhibitor conformation and binding mode. For instance, Khattab et al. [63] showed that the binding interaction of AG1478 with either a kinase important for bacterial drug resistance (APH(3′)-Ia) or a kinase important for pancreatic cancer (MAPK14) can result in a discrete change in AG1478 conformation on a rugged matrix of protein backbone. In this study the authors inferred conformational heterogeneity of the drug AG1478 from excitation spectra, polarity of the AG1478 binding pocket from the emission spectrum and restricted molecular dynamics in the AG1478–kinase complex from excitation-dependent emission spectra. Their fluorescence excitation spectra revealed that the interactions between the AG1478 and APH(3′)-Ia proteins were different from the interactions between the AG1478 and MAPK14 proteins. Emission wavelength-dependent excitation spectra in the former indicates that AG1478 exists in multiple distinguishable and competitive conformations in the APH(3′)-Ia binding pocket. While in the MAPK14 binding site, the degree of conformational heterogeneity of the AG1478 was reduced, as inferred from excitation spectra which were not very dependent on emission wavelength. The fluorescence spectra of the AG1478 were also sensitive to the kinase binding partner, with the AG1478–MAPK14 complex exhibiting a bluer emission than the AG1478–APH complex (Figure 4). The results suggest that AG1478 binds to ATP binding site, which is less polar in MAPK14 than APH, consistent with the analysis of the polarity of the amino acids which constitute the binding sites of the two proteins. Both complexes displayed a red-edge excitation shift (REES) (Figure 4e,f), which the authors attributed to restricted dynamics of the protein atoms around the AG1478 on the fluorescence timescale. Operationally, REES is defined as the red shift in fluorescence spectrum attributed to a shift in the excitation wavelength toward the red edge of the absorption spectrum. REES results from heterogeneity in solvent–fluorophore interactions experienced at the level of individual fluorophores within a distribution. Excitation at the red-edge of the absorbance band preferentially excites those fluorophores with a small energy gap between ground and excited state. While excitation near the absorbance maximum will excite fluorophores whose energy gap between ground and excited states is larger. Provided environmental relaxation after photon absorption is restricted or absent, excitation at the red-edge will therefore give rise to a red-shifted emission relative to excitation at the main absorbance transition. On a different note, the application of REES is not limited to only restricted dynamics of solvent relaxation, rather it can help in identification of various conformations of protein [64], measuring conformational heterogeneity of a fluorophore environment [65]; even dynamic hydration map of protein is envisioned as its future application [66].

More recently, a reappraisal of the red-edge excitation shift data has been made in terms of a model that posits the existence of intermediate or hidden bound states of the AG1478 in the AG1478–kinase complexes [67]. In this interpretation, the red-edge excitation shift is due to discrete but low population states which possess absorption and emission spectra which are progressively red-shifted with respect to the most populated state. This model quantitatively fits the red-edge excitation shift curves and allows extraction of both the number of intermediate states and their associated relative free energies.

Bosutinib [34] is a second-generation dual Abl/Src inhibitor that exhibits potent growth inhibition of CML cells in vitro, is active against multiple Imatinib-resistant BCR-Abl mutations, and has demonstrated efficacy in ongoing clinical trials for Imatinib-resistant chronic myeloid leukemia (CML). Bosutinib is a 4-anilinoquinoline-3-carbonitrile inhibitor that has the similar scaffold as the drugs erlotinib and gefitinib, inhibitors of the epidermal growth factor receptor (EGFR). The details of the interaction between Bosutinib and Abl kinase by fluorescence binding assay. Bosutinib becomes strongly fluorescent (i.e., ~10-fold increase in fluorescence intensity) upon binding to Abl or Src kinases. Both isomers of Bosutinib and protein inhibitor samples showed emission spectra at 480 nm. A titration of Bosutinib with excitation at 280 nm results in quenching of tryptophan fluorescence at 340 nm and a rise in emission at 480 nm, indicating that Förster resonance energy transfer (FRET) occurs between the tryptophan amino acids in the protein and Bosutinib. FRET is the non-radiative transfer of excitation energy from the initially excited donor fluorophore (in this case, tryptophan) to the acceptor fluorophore (in this case the Bosutinib). FRET depends on the orientation of the donor and acceptor moieties, the degree of spectral overlap (i.e., tryptophan emission overlaps with Bosutinib’s absorption) and importantly on separation between donor and acceptor (typical scale is 1–10 nm, FRET depends on the inverse sixth power of separation). The implication is that Bosutinib is located in close proximity to one or more typtophans in the Abl or Src kinases. Here, fluorescence emission intensity was plotted as a function of the Bosutinib concentration and fit to a single binding site model to obtain the equilibrium dissociation constant.

Later, Boxer and Levinson [68] extended their work which highlighted the importance of structured water molecules for inhibitor recognition. They used fluorescence assay to test whether the participation of the drug in the hydrogen bond network affects binding by measuring the dissociation constants of Bosutinib for all Src mutants. From the red-shifts of the nitrile probe of Bosutinib they reported that on average, participation in the hydrogen bond network is responsible for a ~25-fold increase in binding affinity, corresponding to a significant energetic contribution of almost 2 kcal/mol. The key result of their work was to demonstrate that the water-mediated interactions can make a critical contribution to an inhibitor’s ability to distinguish between receptor subtypes, suggesting that greater attention to structured water molecules in drug design may yield inhibitors with improved or altered selectivity properties.

Among few reported fluorescence applications for TKIs, Clayton et al. have employed fluorescence titration analysis to quantify the binding interaction between AG1478 mesylate salt, with and without cyclodextrin carrier, and human serum albumin (HSA) [69]. The increase in fluorescence quantum yield of AG1478 upon binding to albumin was measured at excitation wavelength of 350 nm. The steady-state fluorescence data were tested by a single-site and two-site models. The better fit to the two-site model speculated that AG1478 can interact with at least two different binding sites on albumin molecule. The combined fluorescence and ultracentrifugation data suggested formation of ternary or higher order complexes between AG1478 and carrier protein [69]. Importantly, the mesylate group was found to enhance the affinity of AG1478 for HSA by more than an order of magnitude.

Dömötör et al. [36] investigated binding between small molecule inhibitors (Gefitinib, Erlotinib, Afatinib, Osimertinib, KP2187) and human serum albumin (HSA) using molecular modelling methods and fluorescence spectroscopy. Gefitinib, Erlotinib and KP2187 displayed significant intrinsic fluorescence, which was highly sensitive to solvent polarity and hydrogen bonding strength. In contrast, Afatinib and Osimertinib were non-fluorescent. Accordingly, direct quenching of protein fluorescence and site marker displacement measurements were employed to assess drug binding. Control experiments on the drugs revealed that Erlotinib was in in the neutral form physiological pH, while Gefitinib, Afatinib, Osimertinib, and KP2187 were predominantly in a singly protonated form in the un-complexed state. Fluorescence analysis revealed that all inhibitors bound to site I of HSA (in subdomain IIA) with weak-to-moderate binding affinity (log K = 3.9–4.9). Based on the polarity-dependent fluorescence, the authors were able to assign the polarity of the binding pocket as being mainly hydrophobic with some hydrogen bonding. A second binding site II (subdomain IIIA) was found for Osimertinib and KP2187. The assertions from fluorescence spectroscopy agreed well with modelling of the drug–protein interactions. Studies of the interaction of Erlotinib with bovine serum albumin have found similar binding characteristiscs [70,71].

The research group of James N. Wilson [38] reported the spectroscopic characteristics, drug-macromolecule binding, and fluorescence in-cell imaging of Lapatinib, an EGFR/ERBB-targeting kinase inhibitor. Density functional theory (DFT) calculations disclosed that the 6-furanylquinazoline core of Lapatinib exhibited an excited state with charge transfer character and an S_0_ to S_1_ transition energy of 3.4 eV [38]. Accordingly, optical spectroscopy demonstrated that Lapatinib functions as an environmentally responsive fluorophore, with polarity-sensitive emission. As can be the case for many of these TKIs, the hydrophobicity of Lapatinib renders it insoluble in aqueous solution making it difficult to record the spectrum of the monomeric drug in aqueous solvent. The authors were able to report the spectroscopic properties for the aggregated drug, drug-BSA and drug-ErbB2 kinase domain complexes. Confocal fluorescence microscopy imaging of Lapatinib uptake in ERBB2-overexpressing MCF7 and BT474 cells revealed pools of intracellular inhibitor with emission profiles consistent with aggregated Lapatinib.

#### 3.4.2. Fluorescence Analysis of Drug–DNA Interactions

Therapeutics that are targeted to particular kinases are designed to bind to the kinase domain of the relevant protein. However, there are a number of reports [72,73,74,75,76,77,78,79,80,81,82,83,84,85,86] that these drugs can also bind to DNA, which prompts the question as to whether these drugs may have effects other than just inhibition of kinase activation. In Table 2 we present some selected examples from the literature.

The laboratory of Shi, for example studied the Raf kinase inhibitor Sorafenib [77] and showed that it bound to the minor groove of DNA with a millimolar equilibrium dissociation constant. The EGFR/Her2 inhibitors Gefitinib [79] and Lapatinib [72] were also minor groove binders with slightly enhanced affinity (100 micromolar dissociation constants). Hegde et al. [81] investigated the interaction of Imatinib, anti-leukemia drug with DNA to gain the information about its binding mode on DNA and the conformational change of DNA after binding Imatinib. By competition with acridine orange, Hedge deduced that imatinib inter-collated into DNA with modest affinity (7 × 10^3^ M^−1^).

In these studies [72,73,74,75,76,77,78,79,80,81,82,83,84,85,86] absorbance spectroscopy is normally used to assess binding and to determine binding affinities. Circular dichroism is employed to assess any changes to the conformation of the DNA. Displacement assays with preformed DNA-fluorophore complexes are used to determine binding mode by fluorescence spectroscopy. For example, acridine orange or ethidium bromide are used as inter-collator, and competitive drug binding indicates drug association as a DNA inter-collator While rhodamine B or Hoechst 33288 are minor groove binders and competition with these indicates drug association with the minor groove of DNA.

Fluorescence emission measurements are also useful for the determination of the drug–DNA binding affinities, i.e., the strength of the interaction, especially for drugs that are intrinsically fluorescent. As the first experimental evidence of anilinoquinazoline derivative targeting DNA [86], Goossens and colleagues showed that the weak emission of the free form of a N-methyl analogue of an anti-EGFR inhibitor, EBE-A22 in solution was significantly enhanced when the drug bound to DNA. EBE-A22 exhibited inter-collation binding with an affinity of 2 × 10^4^ M^−1^. Interestingly, EBE-A22 did not inhibit EGFR phosphorylation but was cytotoxic to cells. These studies show that some anti-cancer drugs designed to inhibit kinases can also bind to DNA bit it remains to be seen whether this is a significant event in the cellular or in vivo context.

#### 3.4.3. Fluorescence Analysis of Drug–Lipid Interactions

Among variety of experimental techniques fluorescence has been used to study the interaction of anticancer drugs with membranes using liposomes as mimetic systems. Thus, the partition coefficient of chemotherapeutic drugs (such as Doxorubicin) has been determined by fluorescence quenching [87]. Additionally, fluorescence intensity measurements help in finding membrane drug location and binding [88]. The effects of the packing of ordered domains and in the global order of the membrane were studied by fluorescence intensity measurements [89]. Trummer et al. [71] utilized fluorescence approaches to examine the interaction of Gefitinib with liposomes formed from phospholipid bilayers. Gefitinib complexed with vesicles (egg phosphatidylcholine (ePC):polyethylene glycol (PEG)-distereoylphosphatidylethanolamine (PEG-DSPE) (9:1 mol:mol)) exhibited a fluorescence peak at 380 nm. The emission peak observed for the Gefitinib–bilayer complex was close to that observed for Gefitinib in n-hexane but blue-shifted with respective to the spectrum recorded in ethanol. The authors inferred that the Gefitinib was located near the center of the bilayer in the hydrocarbon region [71].

#### 3.4.4. Fluorescence Analysis of Drug–Cell Interactions

The lipophilicity of the drug, which is somehow related to good transcellular adsorption of the drug, is a major concern in the development of its dosage form, because the drug molecules must penetrate the lipid bilayer of most cellular membranes. Lipophilic groups, such as methyl- or chlorine-, esters enhance diffusion across the blood–brain barrier (BBB) [90], which causes side effects of some anticancer drugs such as Dacomitinib. Besides membrane permeability of pharmacological agents, their sequestration to subcellular compartments is a major determinant of target interaction efficacy and, consequently, resistance. In order to understand the molecular mechanisms underlying drug action and resistance development, Englinger et al. [49] described the fluorescence-based, label-free and organelle-specific intracellular visualization and quantification of the clinically approved multikinase TKI, Nintedanib, in lung cancer cells in vitro as well as in tumor xenoografts of orally treated mice. This study uncovers a central interest in biomedical research, which is the dynamics of intracellular distribution of pharmacological agents, by showing how the intrinsic fluorescence from the drug provides useful information on drug location within cells. It also suggests a mechanism for drug resistance, i.e., the drug appears to be trafficked away from its site of action.

The group of Wilson, using the inherent fluorescence of Lapatinib, was able to image the accumulation, distribution and aggregation of the drug within cells [38]. Confocal fluorescence microscopy imaging of Lapatinib uptake in ERBB2-overexpressing MCF7 and BT474 cells reveals pools of intracellular inhibitor with emission profiles consistent with aggregated Lapatinib. The aggregates observed in some cell studies could be due to a drug deactivation pathway using the lysosomal pathways or on the other hand there is some suggestion that the drug aggregates could be harnessed to provide a controlled-release pharmaceutical.

Trummer et al. [71] utilized two-photon confocal fluorescence microscopy to visualize the distribution of Gefitinib in cancer cells and spheroids formed from rat brain tumor cells. The Gefitinib distribution inMCF7 breast cancer cells (2 µM gefitinib;18 hr incubation) was punctate, intracellular, nucleus-excluded consistent with cytoplasmic vesicles (lysosomes or endosomes). In the 9 L brain tumor spheroids, drug uptake was observed in the outermost layer of cells with limited penetration to the innermost cells. Further work is required to understand the basis for different extent of uptake into these cells. Is this an issue of drug entry into the cell or do the cells with less drug have enhanced efflux mechanisms (such as greater expression of pumps or transporters)? These results invite the possibility of using intrinsic fluorescence imaging to track the long-term dynamics of drugs in complex environments. More work needs to be done in assessing the potential phototoxic or photobleaching of these drugs.

#### 3.4.5. Towards Pre-Clinical In Vivo Models for Cancer Research

Fluorescent anti-cancer drugs have the potential to provide important insights into drug uptake and efflux in tumors implanted in mice. Unfortunately, penetration of the ultraviolet light (used to excite TKIs) into tissue is limited. For this reason, fluorophore-TKI conjugates are used which absorb light in the visible to near-infrared wavelength region, which then enables in vivo monitoring of drug uptake and efflux. (Parenthetically we note that a drug-fluorophore conjugate is not strictly an intrinsically fluorescent drug and thus is beyond the scope of this review). The reader is referred to an excellent recent review [91] of these drug-fluorophore conjugate systems, which in some cases appear to be nearly as good or even as good as the parent (unlabeled drug). One promising development in the area of extrinsically labelled tyrosine kinase inhibitors is in the imaging of tyrosine kinase inhibitor engagement in a model tumor environment within mice. Because receptor levels can vary in different cells, and also because of off-target binding, it is difficult to determine the specific engagement of tyrosine kinase inhibitors. To circumvent these issues, the group of Gibbs applied a paired reagent fluorescence imaging strategy [92]. In this approach, two fluorescently tagged reagents are injected into the mice simultaneously, one specifically targeted to the kinase and the other one untargeted to the kinase domain. By subtracting the non-specific signal from the specific signal, the authors were able to determine the kinetics of specific target engagement in real time in a tumor cell environment. Another area of future development may be in multi-photon absorption. Use of three-photon absorption has been employed to examine fluorophores that normally absorb light at UV wavelengths [93]. If three-photon absorption could be applied to anti-cancer drugs this would enable the visualization of anti-cancer drugs in more complex biological environments without extrinsic labelling.

## 4. Conclusions

The review of a selected series of anti-cancer drugs has revealed that some of them possess optical spectroscopic properties, i.e., absorbance and/or fluorescence, which are sensitive to environment and thus can be used to measure the interaction of drugs with intended targets and non-targets in solution. There is potential to exploit these spectroscopic properties to understand more about the structure (conformation) of the drugs themselves, more about the location of the drugs within the protein environment and more about how dynamics influences the interaction affinities. Can the spectroscopy of drugs reveal subtle differences in binding sites which are missed by other approaches?

The ability to observe the interaction and dynamics of anti-cancer drugs, without the potential interference of needing to add an external label, with targets and (non-targets) in living cells and in animals is an important future goal, which should add the development of more specific kinase inhibitors. If the spectroscopic properties of the drugs change due to binding to a non-target vs. a target, can this be used to assess for drug side-effects or drug resistance? Answers to this question will require translating solution techniques into appropriate biological models.

## Figures and Tables

**Figure 1 biology-11-01135-f001:**
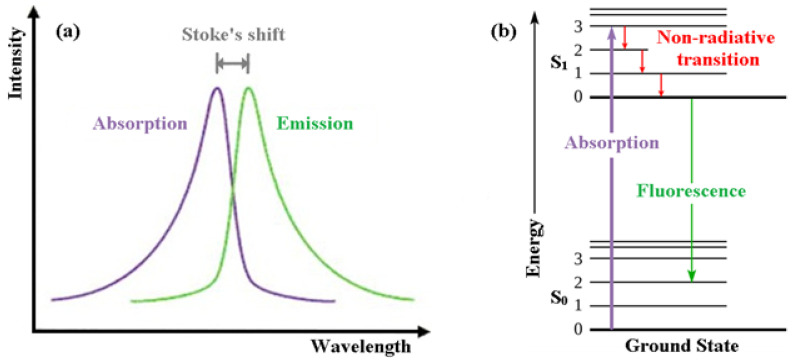
(**a**) Emission intensity vs. wavelength plot. (**b**) Jablonski energy diagram for absorption and fluorescence.

**Figure 2 biology-11-01135-f002:**
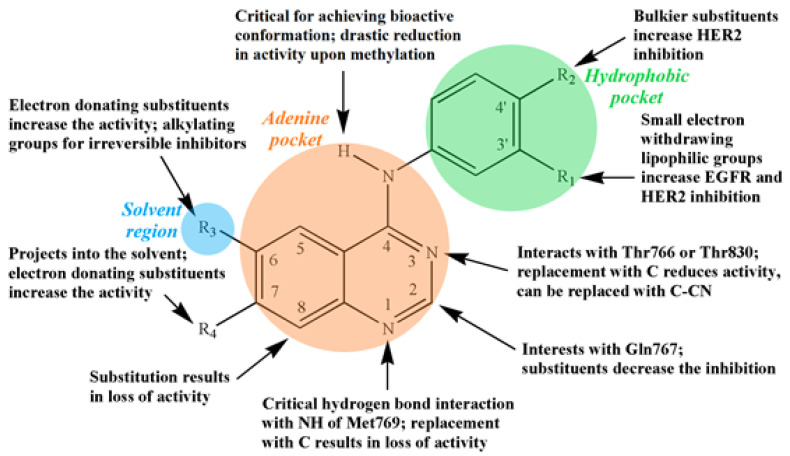
Structural relationship of EGFR inhibition activity for the 4-aminoquinazolines. The hydrophobic pocket (green), adenine pocket (orange) and solvent region (blue) interacts with the N-aryl arm, quinazoline core and tail section, respectively, of the 4-aminoquinazolines. Reprinted/adapted with permission from Ref. [26]. Copyright 2016, Future University.

**Figure 3 biology-11-01135-f003:**
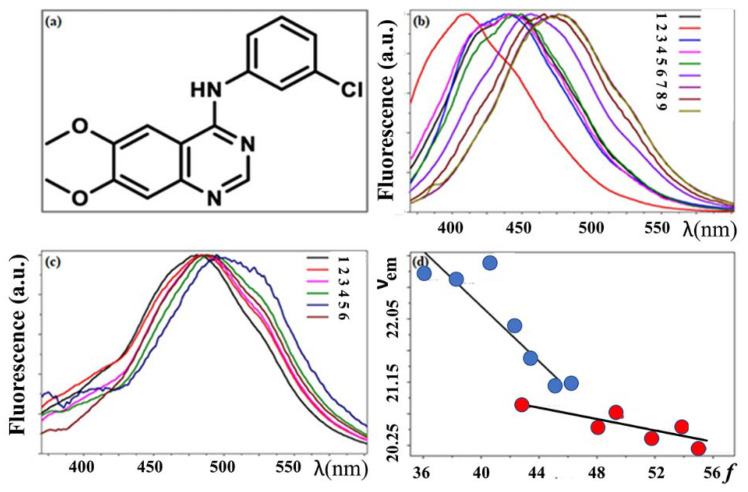
TKI (AG1478) exhibits environmentally sensitive fluorescence. (**a**) Chemical structure of N-(3-chlorophenyl)-6,7-dimethoxyquinazolin-4-amine (AG-1478) (**b**) Fluorescence spectra of AG1478 in aprotic solvents of increasing polarity (λexc = 350 nm, [AG1478] = 3 μM). Note the increased red-shift in solvents with increased polarity. Legend key; 1 = dioxane, 2 = toluene, 3 = choloroform, 4 = ethylacetate, 5 = dichloromethane, 6 = acetone, 7 = acetonitrile, 8 = dimethylformamide, 9 = dimethylsulphoxide (**c**) Fluorescence spectra of AG1478 in protic solvents of increasing polarity (λexc = 350 nm, [AG1478] = 3 μM) Legend key; 1 = tertiary butanol, 2 = 1 butanol, 3 = isopropanol, 4 = ethanol, 5 = methanol, 6 = N, methylformamide. (**d**) The correlation of Reichardt solvent transition energy parameter (***f***) with the emission maxima in wavenumber (10^3^ cm^−1^) for aprotic solvents (blue circles) and protic solvents (red circles) Reprinted and adapted from M. Khattab, F. Wang, A.H.A. Clayton, UV-Vis spectroscopy and solvatochromism of the tyrosine kinase inhibitor AG-1478, Spectrochim. Acta—Part A Mol. Biomol. Spectrosc. 164 (2016) 128–132. doi:10.1016/j.saa.2016.04.009 with permission from Elsevier.

**Figure 4 biology-11-01135-f004:**
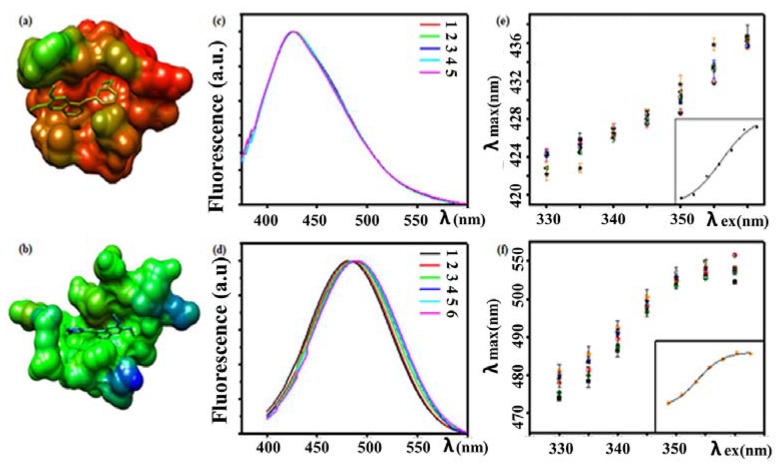
Structure and fluorescence spectra of drug–kinase complexes. (**a**) 3D structural model of an AG1478-derivative in complex with MAPK14. The average B-factor is 18 Å^2^ indicating restricted thermal motion of the atoms in the crystal. (**b**) 3D structural model of an AG1478 in complex with APH(3′)-Ia. The average B-factor is 52 Å^2^ and denotes a less restricted thermal motion (more atomic disorder) of the atoms in the crystal. (**c**) Fluorescence spectra of AG1478–MAPK14 complex recorded at different temperatures (15–37 degrees). Note the wavelength position of fluorescence spectrum is blue-shifted and independent of temperature. Legend key:(1,2,3,4,5) = (15,20,25,30,37) °C. (**d**) Fluorescence spectra of AG1478–APH(3′)-Ia complex recorded at different temperatures (15–37 degrees). Note the fluorescence is red-shifted and displays some thermochromism. Legend key: (1,2,3,4,5,6) = (10,15,20,25,30,37) °C. (**e**) Red-edge excitation shift of the AG1478-MAPK14 complex. Note the progressive red-shift of the emission (y-axis) with increasing excitation wavelength (x-axis). (**f**) Red-edge excitation shift of the AG1478–APH(3′)-Ia complex. Note the progressive red-shift of the emission (*y*-axis) with increasing excitation wavelength (*x*-axis). Reprinted (adapted) with permission from M. Khattab, F. Wang, A.H.A. Clayton, Conformational Plasticity in Tyrosine Kinase Inhibitor-Kinase Interactions Revealed with Fluorescence Spectroscopy and Theoretical Calculations, J. Phys. Chem. B. 122 (2018) 4667–4679. doi:10.1021/acs.jpcb.8b01530. Copyright (2018) American Chemical Society.

**Table 1 biology-11-01135-t001:** Spectroscopic properties of selected kinase inhibitors.

Drug (Intended Target)	Absorbance Max (nm)	Fluorescence Max (nm)
Sorafenib (Raf kinase)	(DCM/ACN) ^1^ 264 nm	487 nm [42,43]
Gefitinib (EGFR)	n-hexane (332 nm, 344 nm)	368 nm, RI ^7^ = 1 [35,36]
Gefitinib (EGFR)	Benzene (332 nm, 340 nm)	446 nm RI = 0.3 [35,36]
Gefitinib (EGFR)	n-octanol (334 nm, 344 nm)	390–450 nm RI = 0.07 [35,36]
Gefitinib (EGFR)	Water (330 nm)	Non-fluorescent
Gefitinib (EGFR)	HSA (not reported)	378 nm [35,36]
Erlotinib (EGFR)	n-hexane (332 nm, 344 nm)	372 nm RI = 1 [36,37]
Erlotinib (EGFR)	Benzene (336 nm, 346 nm)	423 nm RI = 0.38 [36,37]
Erlotinib (EGFR)	n-octanol (336 nm, 346 nm)	400–470 nm RI = 0.07 [36,37]
Erlotinib (EGFR)	Water (332 nm)	Non-fluorescent [36,37]
Erlotinib (EGFR)	HAS ^2^ (336 nm, 347 nm)	380–400 nm [36,37]
Lapatinib (EGFR, HER2)	THF ^3^ (380 nm)	475 nm [38]
Lapatinib (EGFR, HER2)	Methanol (367 nm)	Non-fluorescent
Lapatinib (EGFR, HER2)	BSA ^4^ (361 nm)	423 nm QY ^8^ = 0.07 [38,39]
Lapatinib (EGFR, HER2)	ErBB2 (368 nm)	445 nm QY = 0.30 [38,39]
Lapatinib (EGFR, HER2)	Aggregates (371 nm)	464 nm QY = 0.04 [38,39]
Afatinib	Methanol (246 nm, 340nm)	Not reported [46,47]
Imatinib (Abl kinase)	Water (281 nm)	307nm [40,41]
Dacomitinib (mutated EGFR)	DMSO ^5^ (343 nm)	500 nm ^9^ [44,45]
Nintedanib	PBS ^6^/DMSO (390 nm)	482 nm [48,49]
Bosutinib (Abl kinase)	350 nm	480 nm [34]

^1^ DCM/CAN is dichloromethane/acetontrile; ^2^ HSA is human serum albumin; ^3^ THF is tetrahydrofuran; ^4^ BSA is bovine serum albumin; ^5^ DMSO is dimethylsulphoxide; ^6^ PBS is phosphate-buffered saline; ^7^ RI indicates relative intensity; ^8^ QY = fluorescence quantum yield; ^9^ personal communication MD Lutful Kabir, Swinburne University.

**Table 2 biology-11-01135-t002:** DNA binding properties of some selected drugs.

Drug (Intended Target)	DNA Binding Mode	Binding Affinity (Ref.)
Sorafenib (Raf kinase)	Minor groove	5.6 × 10^3^ M^−1^ [77]
Gefitinib (EGFR)	Minor groove	1.0 × 10^4^ M^−1^ [79]
Lapatinib (EGFR, HER2)	Minor groove	1.0 × 10^4^ M^−1^ [72]
Imatinib (Abl kinase)	Inter-collator	7 × 10^3^ M^−1^ [81]
PD153035 (EGFR)	Inter-collator	Weak [86]
EBE-A22 (cytotoxic)	Inter-collator	2 × 10^4^ M^−1^ [86]

## Data Availability

Not applicable.

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
