# Peer review of "Intrinsically Fluorescent Anti-Cancer Drugs"

_biology, 2022, doi:10.3390/biology11081135_

Round 1

Reviewer 1 Report

In the review paper titled “Intrinsically-fluorescent anti-cancer drugs” by MD. Lutful Kabir, Feng Wang, and Andrew H.A. Clayton submitted to Biology, the authors describe using the intrinsic fluorescence properties of anticancer drugs to study their interactions with biomolecules. While this topic could use a detailed review, the organization of the manuscript is confusing and the topics focused on are not particularly useful in achieving the goal of providing a detailed review on the topic. Additionally, the paper provides detailed explanations of basic principles of fluorescence but goes on to provide very brief explanations of more advanced topics. Overall, this paper should undergo major revisions before reconsideration to address these issues and the ones detailed below.

Questions and Comments:

1.     The introduction really only introduces TKIs without a clear tie to their fluorescence properties. The manuscript would be greatly improved if the introduction introduced the content of the paper as a whole. Additionally, I am not sure a lengthy introduction to TKIs is needed as the uniqueness of this manuscript is the focus on their intrinsic properties and TKIs have been reviewed recently in depth elsewhere including in Nature Reviews Drug Discovery 2021, 20, 551.

2.     Page 3, line 93 section 3 is headed Results but a review paper should not have a results section

3.     Section 3 “Results” has no references cited.

4.     Section 3 “Results,” a similar explanation can be found in most undergraduate textbooks. A focus on the more advanced/unique techniques that exploit the intrinsic fluorescence properties would be more useful.

5.     Page 5, lines 190-197, the concepts discussed here were previously discussed in the “results” section.

6.     Page 6, Table 1, terms used in the field should be used in the column headings. “Absorbance peak” and “Fluorescence peak” should be “excitation maxima” and “emission maxima”, respectively.

7.     Page 6, Table 1, the molar absorptivity and quantum yield should be included for the excitation maxima and emission maxima, respectively, to completely describe their fluorescence properties

8.     Page 6, Table 1, many abbreviations used are not defined such as BSA and PBS.

9.     Page 8, section 3.4, has no references cited

10.  Page 8, section 3.4, a more detailed explanation on how a fluorescence binding assay works would be useful.

11.  Page 8, section 3.4.1, the concepts discussed here were previously discussed in the “results” section.

12.  Page 9, lines 313-318 including a detailed explanation of REES and its utility for studying the interactions of intrinsically fluorescent compounds would be useful.

13.  Page 9, lines 351-352 a more detailed explanation of FRET and its utility for studying the interactions of intrinsically fluorescent compounds would be useful.

14.  Page 11, table 2, references should be cited in numerical style like the rest of the manuscript.

Reviewer 2 Report

In this article, the authors review the studies on structure, interaction, localization of anti-cancer drugs that possess intrinsic fluorophores in them. Applications of fluorescence spectroscopy to structural and dynamical properties of anti-cancer drugs both in isolation and in complexes with proteins, lipids, and DNAs are summarized. The manuscript is well organized and reader-friendly. Before publication, there are just minor points below that the reviewer noticed during the review process.

Minor points:

Please specify several keywords in the first page.

l.56 "are though to" => "are thought to"?

ll.81-82 Which is the more common form with or without a hyphen, "adotrastuzumab" or "ado-trastuzumab"?

l.93 The number of this section is 2, not 3. In addition to this, the title of this section should be changed to more appropriate one.

l.120 "10-8-10-9" should be rewritten with superscripts.

l.129 It would be better to describe the concentration in more common form such as weight percent (w/w % ).

l.131 What is TLC?

l.286 "In another study" requires a reference.

l.351 The first word of the FRET is "Förster" (it requires a tréma).

l.355 is "that" necessary? Perhaps it reads "...which highlighted the structured water molecules..."

l.428 What is the "secondary structure" of DNA? Normally, the secondary structure refers to alpha-helices and beta-strands in proteins. Can one investigate the DNA structure with CD (here does it refer to >250 nm range?)?

l.441 "2x104 M-1" should be rewritten.

l.441 There are two verbs in a row "did not bind inhibit" in one sentence.

In the right-hand column of Table 1, there should be a space between the numerical values and the unit (eg., 487 nm).

The description of Figure 4 (a) and (b) is ambiguous. The authors say that "The red color indicates the restricted thermal motion of atoms (atomic order) in the crystal". Is the structure colorized according to the temperature factor values? Then, the color scale bar should be added to the figure.

Throughout the manuscript, the equation describing the photon energy (the product of the Plank's constant and the frequency) should be written correctly using, for example, the equation editor.

Reviewer 3 Report

This manuscript does a great job of reviewing fluorescent anti-cancer drugs. The authors start with a description of some of the anti-cancer inhibitors and antibodies that target key kinases. They then provide a nice description of the concept of fluorescence before diving into the fluorescent anti-cancer drugs.

I suggest the authors add the importance of fluorescent drugs in cell-culture experiments and other pre-clinical mice models for cancer research

For example, doxorubicin has fluorescent properties and its distribution inside the cells can be assessed using fluorescent microscopy. This is useful because doxorubicin-resistance cells could have a lower intracellular concentration of doxorubicin because of drug efflux. Drugs or gene knockout screens that can reverse such kind of resistance can be discovered by exploiting the fluorescence property of doxorubicin in cells.

Also,  it would be great if we can determine the drug uptake and efflux in a tumor in mice if that is a possibility. There are a variety of ways to exploit the fluorescent properties of anti-cancer drugs in cell-culture and animal models of cancer research and it is important for the authors to include that in the manuscript. This will significantly strengthen the manuscript.

Figure 3(b and c) are blurred out and are hardly visible.

Figure 4 (c-f) are blurred out as well. Please insert high-resolution figures.

Reviewer 4 Report

The review intrinsically fluorescent anti cancer drugs by Kabir et al, does a systemic analysis of intrinsic fluorescent properties of few anticancer drugs and how that information can be utilized to study the binding dynamics of drugs to their targets/off targets. Overall, the authors have done a great job of summing up the information.

Minor concern:

The use of word results (line 93) is misleading as this is a review and should be changed/removed.

Round 2

Reviewer 1 Report

All changes are acceptable

Author Response

Thankyou for finding the changes made to the review acceptable. We appreciate your efforts to help us improve the quality of the review article.